# A Benchmark of Discovering Drug-Target Interaction from Biomedical Literature

**Yutai Hou**[1], **Yingce Xia**[2], **Lijun Wu**[2], **Shufang Xie**[2], **Yang Fan**[3],
**Jinhua Zhu**[3], **Wanxiang Che**[1], **Tao Qin**[2], **Tie-Yan Liu**[2]
[1] Harbin Institute of Technology    [2] Microsoft Research
[3] University of Science and Technology of China
[1] {ythou,car}@ir.hit.edu.cn,    [3] {fyabc,teslazhu}@mail.ustc.edu.cn
[2] {yingce.xia,lijuwu,shufxi,taoqin,tyliu}@microsoft.com

## Abstract

As millions of papers come out every year in the biomedical domain, automatic knowledge discovery (KD) from biomedical literature becomes an urgent demand in the industry. While KD in the biomedical domain attracts much research attention in recent years, the lack of benchmark datasets significantly hinders its progress. In this work, we create a dataset, KD-DTI, for discovering ⟨drug, target, interaction⟩ triplets from literature, which is one of the most important KD tasks in the biomedical domain. KD-DTI contains $14k$ unique biomedical papers, each of which is associated with at least one ⟨drug, target, interaction⟩ triplet. We also provide a semi-supervised dataset with $139k$ unique papers. We present and analyze multiple solutions, including several extractive/generative models and two data enhancement methods. The results show that the performance of those models is far from industry demand, indicating that the dataset presents a challenging research problem for the community. The dataset will be freely accessible after the review process.

## 1  Introduction

Biomedical literature is an important data source for both research organizations and industrial companies to discover knowledge. PubMED, one of the most famous search engines for biomedical literature[1], has indexed more than $30M$ articles, and there are millions of new papers coming out every year [13]. It is impossible to manually check all the papers to obtain useful knowledge. Therefore, it is an urgent demand to automatically discover knowledge from the literature.

The interaction between drugs and targets in human body plays a crucial role in biomedical science and applications [24, 37, 38], e.g., drug discovery, drug repurposing, precision medicine, etc. In biomedical literature, a drug refers to any type of medication, ranging from small molecules like Aspirin, Penicillin to large molecules like Hepatitis B Vaccine. A target could be protein, enzyme or nucleic acid in our body, which binds the drugs we take. Drugs interact with targets in different ways. For example, Aspirin (drug) can inhibit (interaction) COX-1 (target), and Streptokinase (drug) can activate (interaction) Plasminogen (target). For simplicity, we call a triplet of Drug, Target and their Interaction as a "DTI triplet".

Discovering DTI triplets from biomedical papers is challenging. First, lots of terms and aliases (e.g., abbreviations, synonyms) exist in an article, but only a small set of them contributes to DTI triplets, which makes this task harder than conventional relation extraction from general text. As shown in Figure 1, given the title and abstract of a paper, we want to discover the triplet ⟨Clotrimazo, Ergostero,

---

[1] https://pubmed.ncbi.nlm.nih.gov/about/

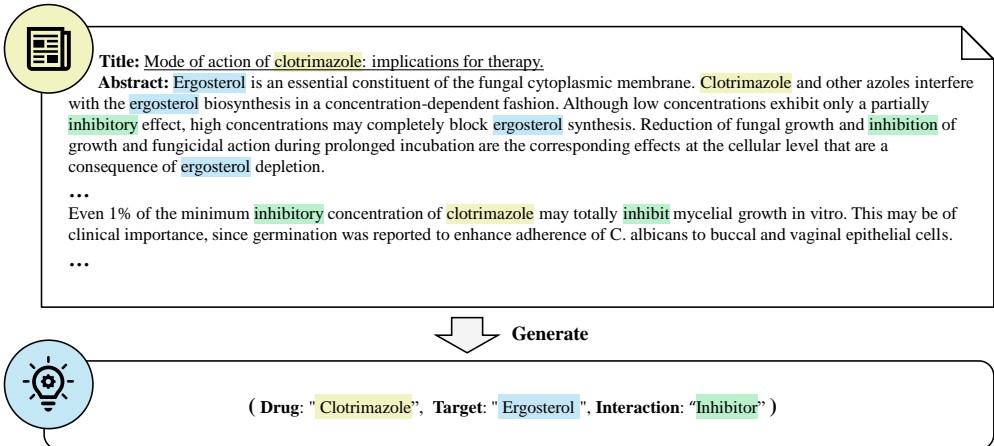

Figure 1: Examples of drug-target-interaction knowledge discovery from literature.

Inhibitor⟩. We can see that there are many terms like "fungal cytoplasmic membrane", "azoles" and "C. albicans", which are not related to the triplet we want to discover and increase the difficulty of the task. Second, there are few public datasets available for this task. Therefore, it is not easy to compare different methods and evaluate the progress on this task.

In this work, we present KD-DTI, a dataset that acts as a starting point of discovering DTI knowledge from biomedical literature. In the dataset, each article is associated with one or multiple ⟨drug, target, interaction⟩ triplets. Due to the diverse term-entity aliases, specialized expressions, and the long document, identifying DTI triplets from papers requires expert knowledge in biomedical domains, as shown in Figure 1. Fortunately, we find that several biomedical databases like DrugBank [35] and Therapeutic Target Database (briefly, TTD) [33] suggest several possible articles from which DTI triplets could be obtained. Based on those databases, we design a scoring mechanism to filter the spurious associations between articles and triplets and keep the remaining ones in the final dataset. We eventually obtain a dataset with $12k$ training samples, $1k$ validation samples, and $1.1k$ test samples. To ensure quality, we manually check all test data.

At last, we explore several possible solutions to DTI discovery, including extractive models and recent generative models, and two data enhancement methods. Experimental results demonstrate that (1) generative models perform better than extractive ones and are more promising for this task; (2) leveraging unlabeled data can further boost the performance of generative models; (3) the performance of all the models is far from industrial demands, even boosted by data enhancement, which suggests that DTI discovery is a challenging task and calls for more research efforts from the machine learning and natural language processing community.

Our contributions are summarized as follows:

(1) We create KD-DTI, the one of first dataset for discovering Drug-Target-Interaction triplets from literature. We expect that such a dataset will boost and advance the research of knowledge discovery from biomedical literature.

(2) We study several baseline methods on the dataset (§4 and §5) and point out future directions for the DTI discovery task (§7).

## 2  The corpus

In this section, we first introduce the acquisition of the dataset (§2.1), and then introduce its statistics and characteristics (§2.2). In order to let readers quickly understand our dataset, we present the data structure of our dataset in Figure 2, where each paper is attached with a list of DTI triples as labels:

```
{
    "pmid value": {
        "pmid": "pmid value",
        "title": "Regulation of ...",
        "abstract": "The effects of treatment ...",
        "triples": [
            {
                "drug": "Drug name or drug id from DrugBank",
                "target": "Target name or target id from DrugBank",
                "interaction": "interaction type"
            },
            … # more triples
        ],
    },
    ... # more samples
}
```

Figure 2: Structure of proposed dataset.

## 2.1 Dataset creation

**Data collection** The DTI triplets in our dataset come from two widely used databases, DrugBank [35] and Therapeutic Target Database (TTD) [33]. (1) DrugBank is a pharmaceutical knowledge base that consists of proprietary authored content describing clinical-level information about drugs.[2] DrugBank covers $14,315$ drugs, $4,885$ targets, $63$ types of interactions and $18,866$ DTI triplets. (2) TTD[3] is a comprehensive collection of various types of drugs, which includes $37,316$ drugs, $3,419$ targets, $109$ interactions and $43,874$ DTI triplets. Given a DTI triplet, if the reference papers are provided and the abstracts of those papers are openly accessible, we record the triplet and the paper. As the first step, we only use the titles and abstracts of the reference papers. For ease of reference, we denote the dataset obtained at this step as $\mathcal{D} = \{D_j, \{Y_{j,k}\}_{k=1}^{K_j}\}$, where (1) $D_j$ is the document (i.e., title and abstract); (2) $Y_{j,k} = (d_{j,k}, t_{j,k}, i_{j,k})$ is the $k$-th triplet of $D_j$, with each element representing drug, target and interaction respectively; (3) $K_j$ is the number of triplets associated with $D_j$.

**Data filtration** As a starting point of structured DTI knowledge discovery, we are only interested in the document which contains enough information to discover a DTI triplet. However, in $\mathcal{D}$, some papers only generally describe some drugs and targets, in which the DTI triplets do not explicitly appear. Therefore, we heuristically filter out the samples in $\mathcal{D}$ by which we cannot obtain the associated DTI triplets. The basic idea is that we require that the drug, target, and interaction in a triplet should be all included in a paper. We describe the details of the filtration process as below.

Given a query $q$ and a document $D$, we first use FuzzyMatch[4] to retrieve all similar words of $q$ and its synonyms in $D$, and denote them as $\mathcal{R}(q, D) = \{r_j\}_{j=1}^{|\mathcal{R}|}$, where $r_j$ is a retrieved phrase. Here the query can be a drug, a target, or an interaction, and both $q$ and $r_j$ could be a single word or a phrase with multiple words. Note that we obtain synonyms of a drug or target from the Drugbank and TTD database, where entities are attached with synonyms. Based on the retrieval results, we categorize $D$ as one of the following patterns for $q$:

1. *Reliable pattern*, where the query and the fetched words are almost the same;

2. *Positive pattern*, where the query and the fetched words share lots of parts in common;

3. *Negative pattern*, where the query is not related to the document.

Detailed patterns are summarized in Appendix B.

---

[2]https://go.drugbank.com/

[3]http://db.idrblab.net/ttd/

[4]An open-sourced tool that leverages Levenshtein distance to fetch similar words to the query. Simple variants are allowed like "+s", "+ed", etc. https://github.com/taleinat/fuzzysearch

| Dataset | # Document | # Relation | # Sentence | # Words | Knowledge |
|---|---|---|---|---|---|
| CPI-DS [7] | N/A | 1 | 2,613 | 486k | Chemical Proteins Relation |
| BC5CDR [14] | 1,500 | 1 | 11,089 | 282k | Chemical Disease Relation |
| ChemProt [2] | 2,432 | 5 | 24,923 | 650k | Chemical Proteins Relation |
| KD-DTI | 14,256 | 66 | 139,810 | 3,671k | Drug Target Interaction |
| KD-DTI (semi) | 139,408 | 66 | 1,556,614 | 39,997k | Drug Target Interaction |

Table 1: Statistic of document level knowledge extraction datasets. "KD-DTI (semi)" denotes semi-supervised dataset in §5.

Denote the matching score between a query $q$ and a document $D$ as $\varphi(q, D)$: If $q$ is a reliable pattern of $D$, we set $\varphi(q, D) = 5$; if $q$ is a positive pattern, we set $\varphi(q, D) = 1$; otherwise, $\varphi(q, D) = -1$. We tried several different score setting and determined the scores with best data quality through manual check. Given any document $D_j$ and its $k$-th DTI triplet $Y_{j,k} = (d_{j,k}, t_{j,k}, i_{j,k})$, the matching score between the triplet and document is calculated as follows:

$$c(D_j, Y_{j,k}) = \varphi(d_{j,k}, D_j) + \varphi(t_{j,k}, D_j) + \varphi(i_{j,k}, D_j).$$

We sort all samples according to the matching scores in descending order, filter out the low-confidence samples whose score is less than zero, and only keep the top $14k$ high-confidence documents-triplets pairs. We pick $1.3k$ documents as the initial test set, the $1k$ as the validation set, and the remaining documents ($12k$) as the training set.

**Human verification** We then manually check all the samples in the test sets. We employ eleven annotators with Ph.D. background. Each (document, DTI triplets) pair is independently checked by two annotators. If their evaluation results are different, another two annotators are involved for discussions. We remove those difficult cases that a consensus is not reached after the discussions of four annotators. We eventually obtain $654$ (document, DTI triplets) pairs from DrugBank, and $505$ pairs from TTD for test set.

### 2.2 Comparisons with previous datasets

Table 1 shows the statistics of our dataset as well as some related datasets. We have the following observations:

(1) In terms of data size (including numbers of documents, sentences and words), our dataset is much larger than previous datasets.

(2) Although ChemProt and CPI-DS also focus on tasks in the biomedical domain, they do not directly serve for drug-target interaction discovery from literature. ChemProt mainly focuses on relation extraction and assumes that entities are given in advance, while our KD-DTI is to discover DTI triplets (instead of relation only) from documents. CPI-DS is to extract relation from single sentences, and thus is much easier than our task that takes long documents as input.

(3) KD-DTI includes a rich variety of relationship types that are not covered by previous datasets. ChemProt covers five relations, and CPI-DS and BC5CDR contain one relation only. In comparison, there are 66 relations in our dataset.

(4) KD-DTI is collected from more than one data sources, i.e., DrugBank and TTD , which could be used to evaluate the generalization or transfer abilities of machine learning algorithms/models.

(4) KD-DTI is collected from multiple data sources, i.e., DrugBank and TTD, which could be used to evaluate the generalization abilities of models.

## 3 Evaluation metrics

We define a set of metrics to evaluate the performance of a model for DTI discovery, covering different granularity: (1) triplet-level metrics, (2) ontology-level metrics, and (3) entity-level metrics.

Let $N$ denote the number of documents/papers in a test set. For the $j$-th test sample/document, the set of its associated DTI triplets is denoted as $Y_j^*$. Let $\hat{Y}_j$ denote the output of a model for the $j$-the sample, which is another set of DTI triplets.

## 3.1 Triplet-level metrics

Following previous work on knowledge extraction [40], we evaluate that given a document, whether the model could correctly discover the corresponding DTI triplets. Since a single paper may contain multiple DTI triplets, we define precision (P), recall (R) and F1 score [45] as follows:

$$P = \frac{1}{N} \sum_{j=1}^{N} \frac{|Y_j^* \cap \hat{Y}_j|}{|\hat{Y}_j|}, \ R = \frac{1}{N} \sum_{j=1}^{N} \frac{|Y_j^* \cap \hat{Y}_j|}{|Y_j^*|}, \ F1 = \frac{2PR}{P + R}.$$

## 3.2 Ontology-level metrics

For industrial applications, given a corpus of documents, one of the important objectives is to find out all possible knowledge from the corpus. To evaluate the knowledge coverage from the corpus level, we define ontology-level metrics (i.e., corpus-level metric) that evaluates how many triplets of the entire corpus are correctly extracted.

Define $Y^* = \cup_{j=1}^{N} Y_j^*$ and $\hat{Y} = \cup_{j=1}^{N} \hat{Y}_j$. The ontology level precision (P), recall (R) and F1 are:

$$P = \frac{|Y^* \cap \hat{Y}|}{|\hat{Y}|}, \ R = \frac{|Y^* \cap \hat{Y}|}{|Y^*|}, \ F1 = \frac{2PR}{P + R}.$$

## 3.3 Entity-level metrics

As mentioned before, a biomedical paper often contains lots of entities, but many of them are not related to the DTI triplets we want to discover. It is important to extract the right drugs, targets, and interactions from literature. Therefore, we assess the accuracy for drugs ($A_d$), targets ($A_t$), interactions ($A_i$) respectively.

Let $D_j^*$ and $\hat{D}_j$ denote the sets of all drugs in the ground-truth triplets and model outputs for the $j$-th sample, and similarly for $T_j^*, \hat{T}_j, I_j^*, \hat{I}_j$. We define drug accuracy, target accuracy, and interaction accuracy as below:

$$A_d = \frac{1}{N} \sum_{j=1}^{N} \frac{|D_j^* \cap \hat{D}_j|}{|D_j^* \cup \hat{D}_j|}, A_t = \frac{1}{N} \sum_{j=1}^{N} \frac{|T_j^* \cap \hat{T}_j|}{|T_j^* \cup \hat{T}_j|}, A_i = \frac{1}{N} \sum_{j=1}^{N} \frac{|I_j^* \cap \hat{I}_j|}{|I_j^* \cup \hat{I}_j|}.$$

# 4 Extractive vs. generative approaches

We explore two types of strategies, the extractive approach (§4.1) and the generative approach (§4.2) for DTI triplet discovery. Experimental results are reported in §4.3.

## 4.1 Extractive approach

For extractive approaches [15, 47, 17], we need to apply named entity recognition (briefly, NER, which is to tag entities), relation extraction (briefly, RE, which is to classify the relations among the discovered entities). We explore two methods: Cascade Relation extraction (CasRel), which is the state-of-the-art extractive method [34] and a pure NER method, which regards relation as a special entity.

**CasRel** CasRel is a cascade tagging method that can jointly perform NER and RE. CasRel leverages BERT to extract representations for input sequences. To find out DTI triplets, CasRel first tags out all possible drugs (i.e., subject) of the input. After that, CasRel searches interactions (i.e, relation) and targets (i.e., objective) for the discovered drugs. For this purpose, we train a classifier for each relation, whose input is the discovered drug and the output is the position of the target, i.e., classifications of

whether each token is the start or end token for the target phrase. The classifier is allowed to output null, indicating that there is no target for this relation.

To use CasRel, we obtain the named entity annotations of drugs and targets by searching the document with `FuzzyMatch`. Although CasRel achieved great success in standard relation extraction tasks like NYT [23] and WebNLG [8], in our setting, the annotations for all entities are automatically obtained without manually check, which limits the performance of CasRel.

**Pure NER method**  In biomedical literature, interactions often explicitly appear in documents with specific forms (e.g., noun, verb, past/present participle). Therefore, it is natural to regard the interaction as a special entity, and use a NER model to figure out the DTI triplets. For this purpose, after obtaining the mentions of drugs, targets and interactions using `FuzzyMatch`, we train a BERT-based NER model where interactions are also types of entity. During training, the BERT-based NER model is trained to predict the possibility of whether a token belongs to entity spans of drugs, targets and interactions. At the inference phase, the trained model tags out token spans of drugs, targets and interactions. For simplification, we choose the drug, target and interaction with the maximum probability (normalized with the length of the BIO representation) to constitute the DTI triplet. With this method, we can predict at most one DTI triplet for each document.

## 4.2    Generative approach

To avoid labeling intermediate annotations (i.e., labels for entities mentions and relations between each pair of entities) and sequentially applying multiple models as extractive methods, we explore generative methods for this task [44, 41]. Specifically, we use a Transformer model [30]. The encoder of Transformer is used to encode the document, and the decoder of Transformer works for generating the DTI triplets. The output of the decoder follows the following format:

$$\texttt{<d>}\ \text{drug}_1\ \texttt{}\ \text{interaction}_1\ \texttt{<t>}\ \text{target}_1\ \texttt{<d>}\ \text{drug}_2\ \texttt{}\ \text{interaction}_2\ \texttt{<t>}\ \text{target}_2 \cdots,$$

where the drug, interaction and target are separated with special tokens `<d>`, `` and `<t>`, and all triplets are concatenated as a longer sequence.

Recently, pre-training achieves great success in NLP areas. We explore two ways of using pre-trained models (The pre-trained model is flexible and we choose both BERT [6] and PubMedBERT [11] models in our experiments.):

(1) **Transformer+BERT** and **Transformer+PubMedBERT**: The encoder of the triplet generator is initialized by the pre-trained models;

(2) **Transformer+BERT-Fuse** and **Transformer+PubMedBERT-Fuse**: Following [49], which successfully incorporates the pre-training models like BERT into sequence generation, we adapt it into our task: In addition to the encoder-decoder based Transformer, we use a pre-trained model like BERT to extract features for the document, which will be fed into both the encoder and decoder of Transformer with attention modules.

## 4.3    Experiments

**Settings** For CasRel, we mainly follow the hyperparameters suggested by [34]. A modification to CasRel is that since our input text can be longer than $512$ (After BPE, there are $762$ abstract longer than $512$ tokens, and $19$ abstract longer than $1k$ tokens), we cut the document into several pieces, each with a length of $512$. We use BERT to encode each piece and concatenate all the representations for further processing. We use $66$ relation-classifiers in total, where each classifier is a single-layer feed-forward network with ReLU activation that taking BERT embedding as input. The drug and target identifier is a single-layer feed-forward network.

For generative models, after tokenization, we apply BPE [26] to both the source sequences and target sequences to reduce vocabularies. We set the number of layers as 2, and the embedding dimension as $256$. We use Adam optimizer with the `inverse_sqrt` scheduler. The learning rate is $5 \times 10^{-4}$ and warm-up steps are $8k$. The dropout and attention dropout of Transformer are set as $0.2$ and $0.1$. The label smoothing is set as $0.2$. The batch size is $12k$ tokens per GPU. For the Transformer with pre-trained models, we explore two methods as introduced in §4.2. We try the conventional $\text{BERT}_{\text{base}}$

| DrugBank | Triplet Level | | | Ontology Level | | | Entity Level (Acc.) | | |
|---|---|---|---|---|---|---|---|---|---|
| | F1 | P | R | F1 | P | R | Drug | Target | Interact |
| CasRel | 15.42 | 13.74 | 17.57 | 18.62 | 20.74 | 16.89 | 27.12 | 23.14 | 31.32 |
| Pure NER | 18.20 | 19.11 | 17.37 | 17.25 | 19.40 | 15.54 | 60.72 | 35.25 | 79.26 |
| Transformer | 27.41 | 28.38 | 26.50 | 25.49 | 26.64 | 24.43 | 53.05 | 52.86 | 79.54 |
| Transformer + BERT | 30.32 | 31.46 | 29.26 | 29.50 | 31.64 | 27.64 | 53.00 | 55.27 | 79.47 |
| Transformer + PubMedBERT | 34.82 | 35.88 | 33.82 | 32.87 | 34.73 | 31.22 | 55.73 | 58.91 | 82.11 |
| Transformer + BERT-Fuse | 34.60 | 35.50 | 33.74 | 33.26 | 35.50 | 33.74 | 56.80 | 55.12 | 79.82 |
| Transformer + PubMedBERT-Fuse | 36.97 | 37.82 | 36.16 | 34.32 | 36.64 | 32.28 | 57.33 | 58.69 | 82.59 |

| TTD | Triplet Level | | | Ontology Level | | | Entity Level (Acc.) | | |
|---|---|---|---|---|---|---|---|---|---|
| | F1 | P | R | F1 | P | R | Drug | Target | Interact |
| CasRel | 5.74 | 4.87 | 7.00 | 6.05 | 9.30 | 4.49 | 18.77 | 18.51 | 18.06 |
| Pure NER | 6.66 | 6.77 | 6.55 | 6.32 | 9.23 | 4.81 | 33.82 | 16.93 | 68.40 |
| Transformer | 6.32 | 6.73 | 5.96 | 5.75 | 6.41 | 5.22 | 10.89 | 56.53 | 87.43 |
| Transformer + BERT | 7.63 | 7.87 | 7.41 | 7.36 | 8.44 | 6.53 | 14.44 | 51.98 | 87.72 |
| Transformer + PubMedBERT | 7.81 | 8.28 | 7.41 | 7.11 | 7.83 | 6.52 | 12.83 | 58.47 | 86.93 |
| Transformer + BERT-Fuse | 8.34 | 8.42 | 8.27 | 7.59 | 8.14 | 7.10 | 15.46 | 53.37 | 87.03 |
| Transformer + PubMedBERT-Fuse | 8.88 | 9.21 | 8.57 | 7.87 | 8.83 | 7.10 | 14.60 | 61.97 | 89.50 |

Table 2: Results of the document to triplet discovery on DrugBank and TTD. "CasRel" and "Pure NER" are two extractive methods leveraging BERT, and the remaining are generative ones. "-Fuse" denote using pre-trained language models in the fusing manner following [49],

model and PubMedBERT$_{\text{base}}$ model, in which PubMedBERT$_{\text{base}}$ is trained using abstracts of all PubMed papers. All models are trained on a single V100 GPU.

**Results and analysis:** The test results of DrugBank and TTD are reported in Table 2. Due space limits, we leave the standard deviation of results in Appendix D and the case study in Appendix E. We have the following observations:

(1) Generative methods obtain better results than the extractive method (i.e., CasREL) on KD-DTI, in terms of triplet-level metric and ontology-level metric. One reason is that our task lacks manual annotation of intermediate labels such as the BIO representations of all entities and relations among any two entities. We obtain such intermediate labels with FuzzyMatch, which are usually of poor quality and therefore impair performance of extractive methods. For DTI triplet discovery task, such intermediate labels are often hard to obtain, and we should keep exploring how to improve performances without intermediate labels.

(2) For extractive methods, the pure NER method outperforms CasRel on triplet-level metric and entity-level metric. Specifically, for entity-level drug accuracy, the pure NER method even achieves the second best result. This shows when intermediate labels are lacking and the relations among entities are comprehensive, simplifying this problem (like extracting only one triplet for a document) is another choice.

(3) Using pre-trained models is helpful for our task. Taking DrugBank as an example, for triplet-level F1, after using conventional BERT to initialize the encoder, the metric can be improved from $27.41$ to $30.32$. After using PubMedBERT, which is a model pre-trained on all abstracts of PubMed, we achieve an even higher F1 score, $34.82$. This demonstrates the effectiveness of pre-training, especially in-domain pre-training.

(4) The manner of using pre-trained models also matters. Comparing with directly initializing the encoder with a pre-trained model, we find that fusing pre-trained language model following [49] can further boost the performance: "BERT-Fuse" and "PubMedBERT-Fuse" obtain more than $4$ and $2$ point improvement over BERT and PubMedBERT respectively.

(5) The scores on TTD are lower than DrugBank because TTD is a harder dataset. To verify this, we calculate the minimal distance between drugs and targets: Given a document $D$ and a DTI triplet $(d, t, i)$, let $P_d$ and $P_t$ denote two sets which are positions of drugs and targets obtained by FuzzyMatch in $D$. The distance is defined as $\min_{p_d \in P_d, p_t \in P_t} |p_d - p_t|$. For DrugBank and TTD,

| | Triplet Order | D-I-T | D-T-I | I-D-T | I-T-D | T-I-D | T-D-I |
|---|---|---|---|---|---|---|---|
| **DrugBank** | Transformer | **27.41** | 26.34 | 26.66 | 25.15 | 25.38 | 25.18 |
| | Transformer + PubMedBERT-Fuse | **36.97** | 36.13 | 32.48 | 33.75 | 34.89 | 36.54 |
| **TTD** | Transformer | **6.32** | 5.01 | 5.87 | 4.72 | 5.81 | 4.37 |
| | Transformer + PubMedBERT-Fuse | **8.88** | 8.52 | 7.83 | 7.28 | 8.42 | 7.12 |

Table 3: Results of the generation with different triplet orders.

the average minimal distances over all test samples are $34$ and $51$, which shows that identifying the DTI triplet from TTD requires understanding a longer document.

(6) While using pre-trained models achieves the best results, we observe that it suffers from overfitting: The F1 score on the training set is $70.29$ for PubMedBERT, which is much higher than those on the validation set ($23.33$) and test set ($22.9$, the average score of two test sets). We find that simply using larger dropout or label smoothing does not help, which suggests better regularization techniques are needed for this task. More details are in Appendix C.

**Effect of generation order.** As mentioned before, we learn to generate drug-target-interaction triplets sequentially for generative methods. An advantage of this method is that we could leverage the dependency among the triplets to improve the generation quality. A question arises: does the order of elements in DTI triplet matter? To find it out, we enumerate all six orders of the triplet on the standard Transformer model and the PubMedBERT-fused model. The results are in Table 3. Generally, the order of (drug, interaction, target) performs better, indicating that the order of triplet should be consistent with natural language order (i.e., subject-verb-object).

## 5 Data enhancement

As shown in the previous section, our dataset is not very large in terms of training data, and thus pre-trained models (e.g., PubMedBERT) helps a lot by using unlabeled data. In this section, we explore two data enhancement methods to leverage unlabeled data: distance supervision [16] and knowledge distillation. We first introduce how we collect and filter the unlabeled data, followed by the description of the two methods, and finally report the results.

We download abstracts of indexed by PubMed. For each document (i.e., title and abstract), we use ScispaCy [20], an open-sourced NER tool to find out all possible drug and target entities, and use `FuzzyMatch` to find out all possible interactions included in KD-DTI. By doing so, we collect a set of ⟨drug, target, interaction⟩ triplets extracted from those documents/abstracts. We then count the numbers of occurrences of each DTI triplet across all documents, and delete DTI triplets with less than $10$ occurrences.[5] We keep the documents that have at least one DTI triplet after the deletion. We eventually obtain a dataset with $139k$ documents, denoted as $\mathcal{D}_{\text{semi}}$, and we will also release it. We call the triplets in this dataset "pseudo" triplets, since they may be noisy. Next we describe two methods to filter out low-quality data from $\mathcal{D}_{\text{semi}}$.

### 5.1 Two data enhancement methods

**Distance supervision**: Given any DTI triplet $(d, t, i)$ in KD-DTI, we use `FuzzyMatch` to search all $\bar{D}_j$ in $\mathcal{D}_{\text{semi}}$. If we find reliable patterns or positive patterns of both $d$ and $t$, we assign a pseudo label/triplet $(d, t, i)$ to $\bar{D}_j$. Denote the obtained dataset as $\mathcal{D}_{\text{DS}}$, which has $15k$ samples.

**Knowledge distillation**: We use a pre-trained Transformer model to generate DTI triplets for each document.[6] If the Transformer model does not generate any triplet for a document from $\mathcal{D}_{\text{semi}}$, we remove such a document from $\mathcal{D}_{\text{semi}}$. Each remaining document in $\mathcal{D}_{\text{semi}}$ is associated with at least one generated triplet and at least one pseudo triplet. If at least two elements (e.g., drug-target,

---

[5]According to our preliminary exploration, if we randomly select a drug, a target and an interaction from our dataset, most of those DTI triplets occur less than 4 times in all the PubMed papers.

[6]For simplicity, we use the "Transformer" model without BERT in Section 4.3. We will explore more advanced models in the future.

| DrugBank | Triplet Level | | | Ontology Level | | |
|---|---|---|---|---|---|---|
| | No Enhance | + DS | + KD | No Enhance | + DS | + KD |
| Transformer | 27.41 | 29.92 | 30.57 | 25.49 | 27.26 | 28.04 |
| Transformer + PubMedBERT-Fuse | 36.97 | 35.11 | 39.78 | 34.32 | 35.15 | 38.87 |

| TTD | Triplet Level | | | Ontology Level | | |
|---|---|---|---|---|---|---|
| | No Enhance | + DS | + KD | No Enhance | + DS | + KD |
| Transformer | 6.32 | 6.99 | 7.20 | 5.75 | 6.19 | 6.66 |
| Transformer + PubMedBERT-Fuse | 8.88 | 10.83 | 11.27 | 7.87 | 8.01 | 9.64 |

Table 4: Comparison of data enhanced methods.

drug-interaction, or target-interaction) of a pseudo triplet are the same as those of a generated triplet, we keep this document; otherwise, we delete it. After filtration, there are $5.8k$ documents left in the dataset. Denote this dataset as $\mathcal{D}_{\mathrm{KD}}$. Note we will use the pseudo triplets in $\mathcal{D}_{\mathrm{KD}}$ for the following experiments; the generated triplets are only used for filtration, but not for model training.

## 5.2 Results

As generative models perform better than extractive ones, we focus on generative ones in this sub section and conduct experiments with Transformer model and Transformer + PubMedBERT-fused model. We merge the KD-DTI corpus with $\mathcal{D}_{\mathrm{DS}}$ and $\mathcal{D}_{\mathrm{KD}}$ respectively to get two enlarged datasets, and then train models on them. Instead of training from scratch, we find that initializing the parameters from a model trained on the parallel corpus KD-DTI is better.

The results are shown in Table 4. We have the following observations:

(1) Enhanced with $\mathcal{D}_{\mathrm{KD}}$, we achieve more than two point improvement on DrugBank, for both Transformer and Transformer + PubMedBERT; On TTD, significant improvements are also observed.

(2) Enhanced with $\mathcal{D}_{\mathrm{DS}}$, the generation performance is also generally improved, but not as much as $\mathcal{D}_{\mathrm{KD}}$, which shows that the quality of the synthetic data is not as good as that from knowledge distillation. This is consistent with the discovery in [40]. Our conjecture is that the documents are rich of entities and noises, and simply using distance supervision without a scoring mechanism cannot lead to significant improvement, especially when the model equips pre-trained knowledge.

From observation (1) and (2), we can also conclude that pre-training and assigning pseudo labels to the unlabeled data are two orthogonal ways, both of which deserve more attention in the future.

(3) We also directly combine $\mathcal{D}_{\mathrm{semi}}$ with the parallel KD-DTI dataset (which is up sampled by five times) and get the largest training dataset in our experiments. However, while training Transformer (without BERT) with this large dataset, the triplet-level F1 scores on DrugBank and TTD are $18.19$ and $3.15$ respectively, which are much worse than training on KD-DTI only. This demonstrates the necessity of quality control in data enhancement.

(4) Even if data enhancement can boost DTI discovery, the overall accuracy is still not very high. For example, the triplet-level F1 on TTD is less than 11.2. That is, DTI discovery is a challenging task. We need to design better models, algorithms, and/or data enhancement methods to meet the expectation of real-world applications.

## 6 Related work

Early research efforts on knowledge discovery focus on discovering knowledge within single sentences [43, 18, 1, 46]. However, lots of knowledge are expressed by multiple sentences [31, 40]. Therefore, document level knowledge discovery is explored, where the existing solutions are often graph-based methods [22, 21, 31, 4, 19] and pre-trained language model based methods [5, 29, 32, 12].

When comes to biomedical knowledge discovering, previous work on this task mainly focus on mining knowledge on large, unstructured, and unsupervised data [39, 48, 27, 42, 9, 28, 10]. Unlike us,

most of these works do not directly extract knowledge triples from papers. [25] propose to discover knowledge from knowledge graph, while we directly discover knowledge from paper text. [3] focus on predicting the relations between bio-concepts and disease. For discovering knowledge triplets from literature, existing works attempt to generate the relationships between disease and genes and targets, e.g., GDA [36] and BC5CDR [14]. Note GDA is a pure weakly-supervised data without direct human supervision. The genes, diseases, and chemical substances in those work are easier to recognize, and the extracted relationships are relatively simple (only one relation type) Different from them, our dataset covers much more diverse entity terms and more relations. ChemProt [2] and CPI-DS [7] are two related datasets that are about to discover chemical proteins relation on document-level and sentence-level respectively. However, both of the two datasets mainly focus on relation extraction and the entities are given in advance, while our KD-DTI is about to jointly discover the DTI triplets from the document. On the other hand, our datasets have more target and relational types and are much larger in volume than existing datasets.

# 7 Conclusions and future directions

In this work, we have created the first dataset, KD-DTI, for discovering ⟨drug, target, interaction⟩ triplets from biomedical literature, which is one of the most important knowledge discovery tasks in the biomedical domain. We hope this dataset will boost and advance the research for this task.

There are multiple directions to explore, based on this dataset and to improve it.

(1) *Accuracy improvement*: We have shown that the performance of several state-of-the-art models is still far from industry demand. Therefore, how to improve accuracy for the task is an important research problem. As shown in this paper, designing better generative models and combing with pre-trained models properly are promising directions. How to effectively leverage unlabeled data (beyond pre-training) is also worthy of exploration. In addition, we should propose more effective regularization techniques to improve the generalization abilities of DTI models.

(2) *Dataset improvement*: We have created the first dataset for the DTI discovery task. The dataset can be improved in terms of scale and quality. Furthermore, there are many other knowledge discovery tasks in the biomedical domain, which also need public datasets for algorithm evaluation and fair comparison.

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
