# OpenReview forum: "A Benchmark of Discovering Drug-Target Interaction from Biomedical Literature"
_NeurIPS.cc/2021/Track/Datasets_and_Benchmarks/Round1 — Submitted to NeurIPS 2021 Datasets and Benchmarks Track (Round 1)_

### Official Review · Reviewer_TXfD · 2021-07-01
**Interesting paper for KD in the biomedical domain with hand-validated dataset, but that can benefit from clearer writing**

**Rating:** 7
**Confidence:** 4

**Strengths:**

- Difficult task indeed
- Rather large dataset
- Human validation effort is significant
- Nice summary of results in page 7

I think this dataset is a step forward in the task of KD in the biomedical domain, relevant to ML researchers in this field and not only.



**Weaknesses:**

* The *biggest* weakness of this paper is that the authors assume a reader is very familiar with many different concepts. While reading this paper the first time, I have made many assumptions (as I was missing data and context that SHOULD have been there from the start), only to have them partially clarified in the rest of the paper, and most after a second and third re-read.

There are a few things that you have to clarify from the start:

1. Please give a visual example of how a DTI triplet in your dataset looks like. Even if you provide the annex with details and the dataset for reviewers, for people reading the paper for the first time, it should be very clear on what data you offer, and a first read throughout the paper does not reveal this. I see that the format is in the annex, but I would much rather have it as a figure on the side of the first or second page.

For example, following my comment above, in table 1, I can't immediately see if the #Sentence refers to the totality of sentences in the dataset, or only for annotated sentences? Only later did I clarify that you do not annotate sentences at all.

2. Explain why you can't annotate/link DTI entities to sentences. You only give clues about this that pop-out later or on a second re-read, like: on line 211 you say: "our task lacks intermediate labels such as the BIO representations", or only when you get to line 300 do you say " However, lots of knowledge are expressed by multiple sentences [31,41]. Therefore, document level knowledge discovery is explored" which points at why you did not mark sentences. This is clearly obvious to you, but for me, working with datasets like SQUAD, XQUAD, where the target is marked in the input, I'm left wondering why, if you identify drugs and targets with fuzzy match, you don't mark them in the abstracts you identified them from. Please add a paragraph where you make this obvious by explaining the reasons behind - it helps a low with readability.

There are other, less critical issues:

3. Levenshtein for matching synonyms of drugs and targets? Shouldn't this be better matched with exact matches? For example there are several open sources for this: e.g 1st page on google gives you chemspider.com where Aspirin is "1,2-dihydroxybenzene" and has a synonym "AIBN", as well as its chemical formula. I'm thinking that a distance 1 value for Levenshtein for Methane and Ethane will give you a positive pattern even though they are different compounds. Or if if you use Levenshtein only of training "s" and "ed", explain why please.

4.  The Pure NER method is not well explained. First you say you use fuzzy match to select drugs and targets, then jump to inference of a BERT model. Please clarify what you do in the training phase and what you do at inference. What exactly do you mean by "we choose the drug, target and interaction with the maximum probability", given that you say you choose the drug and target by fuzzy match? Although I think I understand what you actually do, for other readers this will probably result in the same confusion I had when I first read this paragraph. I know space is limited, but rather than a confusing paragraph, better say a one-liner and point to a bigger explanation in the annex, or at least rewrite in a way that makes sense.

5.  Line 194 : "since our input document is extremely long, we cut the document into several pieces, each with a length of 512." Abstracts are usually less than 2-300 words long and shouldn't fit in more than 1, 2 or maybe 3 BERT contexts of size 512. Could you clarify what "extremely long" and "several" means? By reading this I'm wondering if you're not referring to abstracts here, but to the actual full paper itself.

6. Please specify how much time the human evaluators spent checking out the 14k dataset. Please be honest.

**Additional Feedback:**

While the subject of the paper is good, I think the paper could have been much better / clearly written.

In my opinion, a dataset paper should focus more on the dataset and less on the experiments, even to the point of moving experiment details fully in the annex, and only having a max 1-2 pages out of 8 dedicated to experiment results only. This would allow you much more space to *clearly* define the dataset, as the dataset is what is of actual value.

Overall, the main reason for my rating is the hand-validation work done, as I know that is a big effort. Otherwise, I would have rated the paper lower, as it is of average clarity on organization, low on relation to prior work, too much space dedicated on experiments that are unreproducible (you only say you'll release the dataset not the experiment scripts, which is a poor choice), and, most importantly, makes it difficult for a first-time reader to *quickly* understand how he/she could use the dataset in his/her work.


Update 20-07-21:

I have read the updated paper and the authors' replies, and I acknowledge that the paper is in better shape than before. However, not all things I have argued about have been clarified (and I understand that is difficult, and that the paper would need a significant rewrite for this). I upgraded my rating to 7.

**Clarity:**

No, the paper can be improved, please see the first points in the Weaknesses section.

**Correctness:**

Discovering knowledge is a difficult task, so the dataset is an asset in any way it is built, considering that it is hand-validated.

I think that using Levenshtein to fuzzy match, and to heuristically assign values (5, 1, -1) to sets of triples is a rather weak method, but hey, if it works (here read: "human validated") it's not wrong. Especially since there are no better simple methods, at least for the time being.

Given that the main contribution of the paper is the introduction of the *dataset*, the experiments done on it (the KD discovery methods using NER, BERT, etc.) are just some baselines that are meant to give the reader a baseline of the task and not be SOTA (or resemble SOTA) in any way. Thus, I have no major objections here, as a model on this dataset should be the subject of its own paper.


**Documentation:**

I think the paper lacks high-quality documentation. Also, the authors specify that they don't release the data until after the submission acceptance. This is an odd choice in my opinion. For me, what would be of high value, would be a clear site / github / document / script etc. that would guide me to download and use the data directly.

On the flip-side I understand that licensing with the TTD is a manual step that must be done prior to getting the data.

However, the supplemental material put together (the pdf with a few screenshots and a few lines of text) feels a last-minute put-together of the steps to actually get and use the dataset.

**Ethics:**

I don't think there is anything about ethics in this dataset.

**Relation To Prior Work:**

It is mentioned only briefly, and the related work is small.

Please *do not* give a list of 9 papers like: " and unsupervised data [39,49,27,43,9,28,10,3,25]. ". I don't think there's a single person ever that has actually opened nine papers on a generic subject. Also this feels copy-pasted there just to fill up the references section.

Just to give you an idea, there are 28 references in 14 lines of text. I haven't seen many other articles that have this density of citation per number of lines. You have 50 references in total. Also reference 40 is identical to 41, something that gives me even more the feeling that it's just copy-paste from other articles.


**Summary And Contributions:**

The paper proposes a new dataset that links triples (drug -> interaction -> target) to article titles and abstacts where each interaction can be discovered (knowledge discovery).
The authors offer a 14k hand-validated triples dataset, as well as an order of magnitude larger semi-supervised one.
Also, there are a number of experiments that prove the difficulty of the task.

---

> ### Author Response · Authors · 2021-07-14
> **Thanks for the recognition and suggestion of our work.**
>
> 1. "Please give a visual example and add explanations of sentence-level annotation and the Pure NER method."
>
> We will follow the suggestion and rearrange the presentation. Please check Figure 2 of the new draft.
>
> 2. "Explaining the reasons behind lacking intermediate labels."
>
> Sorry for the unclear description.  Lacking intermediate labels means lacking high-quality manual annotations for intermediate labels and we actually train models with fuzzy-matched labels as mentioned by the reviewer.
> The description is updated in Line217-221 of the new draft.
>
> 3. "Levenshtein for matching synonym"
>
> As we mentioned in Line 82, we deal with alias by searching both target and its synonym in the fuzzy match process.
> To find out synonyms of a target, we obtain a synonym table from DrugBank and TTD database, where each entity is attached with synonyms.
>
> 4. "The Pure NER method is not well explained."
>
> We update and enrich the descriptions of the Pure NER method in the new draft.
>
> 5. "Could you clarify what 'extremely long' and 'several' means?"
>
> After BPE, there are 762 abstracts longer than 512 tokens, and 19 abstracts longer than 1k tokens. The related description is updated at Line198 of the new draft.
>
> 6. "Please specify how much time the human evaluators spent"
>
> We required the annotator to record the annotation time, and each paper takes about 5-10 minutes for a single annotator.
> The entire annotation process took two weeks in total.
>
> 7. "Release the dataset, not the experiment scripts, which is a poor choice. "
>
> We will release all experiment scripts not only the dataset, which is currently updated at: https://github.com/bert-nmt/BERT-DTI.
> We will also maintain the dataset in the same repositories.
>
> 8. " It is odd to release the data after the submission acceptance."
>
> Although the entire dataset has obtained the necessary license, it is still in the process of our internal review.
>
> 9. About "Relation to Prior Work"
>
> We briefly described the related work due to space limitations.
> Following the suggestions, we updated the Related Work in the new draft.

---

### Official Review · Reviewer_k9Y2 · 2021-07-01
**Strong benchmark experiment but questionable data quality, more like a main proceeding paper**

**Rating:** 5
**Confidence:** 5

**Strengths:**

The author conducted comprehensive experiments on their dataset.

The task of extracting knowledge from biomedical literature is valuable.


**Weaknesses:**

The dataset is not very novel. It is a filtered version of the union of DrugBank and TTD. The scoring mechanism seems simple and basically just filters the ones that don't contain the biomedical entity in the abstract. The advantage of KD-DTI compared to the union of DrugBank and TTD is not clear.

The dataset is not well described. I would like to see a full list of interaction types, statistics of the biomedical entities and interactions, the source of the data, etc. Some questions: How do the authors treat aliases of the same biomedical entities? Is there a naming standard for biomedical entities in their dataset? How do the authors treat the hierarchy of the biomedical entities? Such as, in Fig1, The first sentence says statins are inhibitors of HMG-CoA reductase, which itself is a piece of valuable knowledge. The literature further lists 5 specific types of statin, and the DTI extracted is one of the 5. Does the dataset only include specific entities or may include higher-level class names?

The data only include titles and abstracts, which is incomplete in terms of literature.

Some flaws in domain knowledge make the quality of manual checking questionable.


**Additional Feedback:**

This paper should cite the two data sources: https://pubmed.ncbi.nlm.nih.gov/29126136/, https://pubmed.ncbi.nlm.nih.gov/31691823/

The calculation of the ontology metric is not clear enough.


**Clarity:**

The description of the dataset is not sufficient. More statistics in terms of the DTI, such as the distribution of the biomedical entities, distribution of interactions is necessary.
Acronyms are not spelled out.
The metrics are timed 100?


**Correctness:**

There are not enough descriptions of the data. Apart from that, the baseline results and conclusion seem reasonable.
There are flaws in the introduction of the task:

1. Line 26-27, enzymes are a subclass of proteins.

2. Line 27, the author mentioned streptokinase as a drug that activates plasminogen, parallel to aspirin inhibits COX-1. But streptokinase is a kinase (enzyme) that activates the following chemical reaction (here is the hydrolysis of plasminogen). Thus the true relation is that plasminogen is a substrate of streptokinase. An activation interaction of a drug molecule (opposite of inhibition) enhances the biological activity by binding to the target, such as isoproterenol. Those ligands are usually called agonists (https://en.wikipedia.org/wiki/Agonist).

3. Figure 1 shows an example of extracting the desired triplet: <Rosuvasta, HMG-CoA reductase, Inhibitor> from literature. However, the literature presents 5 triplets: pravastatin, fluvastatin, cerivastatin, atorvastatin, and rosuvastatin are inhibitors of HMG-CoA reductase. The author only extracted one for rosuvastatin and mentioned pravastatin and fluvastatin are irrelevant, which is not correct. The second paragraph in Figure 1 says rosuvastatin is the only one whose binding is enthalpy-driven among the five, this doesn’t mean the other four don’t bind. This simply means their binding is entropy-driven.



**Documentation:**

There are enough details on data collection. As for the benchmark, the data enhancement part is not detailed enough, others are good.

**Ethics:**

No.

**Relation To Prior Work:**

I think the author should compare the KD-DTI with DrugBank and TTD in terms of data quality. As most entries in DrugBank and TTD provide a reference to the original paper, gathering the abstract of those papers doesn’t seem like a huge advantage.

Also, the author mentioned ChemProt is designed for extracting relation given biomedical entities, then it should be qualified training data for DTI extraction task. A comparison between ChemProt and DK-DTI might be helpful.


**Summary And Contributions:**

The authors propose a scoring mechanism to filter the spurious associations between articles and drug-target interaction (DTI) triplets. They used the scoring mechanism to filter DrugBank and Therapeutic Target Database (TTD) to obtain a dataset (KD-DTI) and studied several baseline methods on it.

---

> ### Author Response · Authors · 2021-07-14
> **The proposed benchmark mainly aims to boost the task of discovering DTI knowledge from literature text, rather than simply filtering DTI triples presented in databases.**
>
> 1. "The advantage of KD-DTI compared to the union of DrugBank and TTD is not clear."
>
> (1) In DrugBank and TTD, there is no explicit evidence that a DTI triplet can be mined from a given abstract, which is the cornerstone of knowledge extraction. We provide the abstract-DTI mappings with both manual and automatic annotations.
>
> (2) We would like to emphasize that our motivation is to build a dataset for discovering the drug-target-interaction triplet from the literature. In comparison, DrugBank and TTD are two datasets storing the facts in medicine, not about mining new knowledge from literature. This is the main difference.
>
> 2. "The dataset is not well described."
>
> We update the paper with the required information. Please check Figure 2 in the Appendix of the new draft. Statistics of biomedical entities require the classification of entities, and there is not enough time to do this during the rebuttal. We will add it in a later version.
>
> 3. "How do the authors treat aliases of the same biomedical entities? "
>
> As we mentioned in Line 82 and in Line 86 of the new draft, we deal with alias by searching both target/drug and its synonym in the fuzzy match process.
> To find out synonyms of a target or drug, we obtain a synonym table from DrugBank and TTD database, where each entity is attached with synonyms.
>
> 4. "How do the authors treat the hierarchy of the biomedical entities?"
>
> Given an abstract and a drug-target-interaction triplet, we keep the name as it in DrugBank or TTD. We do not specifically introduce the hierarchy of the biomedical entities. Our current task is to discover knowledge from the literature and leave the refinement in future work.
>
> 5. "The data only include titles and abstracts"
>
> As an early attempt at this task, we follow previous work (ChemProt and GDA) and explore extraction on abstract and title, which already include core information of a paper. (See Line71)
> Extraction on full text could be an important future step.
>
> 6. About correctness.
>
> (1) All knowledge is provided by DrugBank and TTD, which can provide the necessary guarantee for the correctness of the data.
> For the example of the pointed "streptokinase as a drug that activates plasminogen", we obtain the information in DrugBank at the page of streptokinase:  https://go.drugbank.com/drugs/DB00086#targets
>
> (2) For the sake of rigor, we have updated all the correctness issues in the introduction of the new draft.
>
> 7. The calculation of the ontology metric is not clear enough.
>
> We have enriched the description of the ontology metric in the new draft.
>
> 8. "Cite the two data sources."
>
> We cited the data source of DrugBank and TTD in Line64. The suggested new version of DrugBank reference is now updated in the new draft.

---

### Official Review · Reviewer_oYKc · 2021-07-04
**Interesting data, hindered by lack of documentation or plan for maintanence**

**Rating:** 4
**Confidence:** 1
**Correctness:** The claims seem fine to me.
**Clarity:** I thought the paper was clear.

**Strengths:**

The paper build a large dataset and manually annotates a significant subset. This dataset is much larger and richer than the others the authors identify.

Drug interaction extraction is an important task, and setting baseline benchmarks may improve the field.

**Weaknesses:**

I can't find a link to the data or code for reproducing the work in the paper. The attached supplementary files are just shell scripts that lack documentation or explanation.

The authors don't mention how the benchmark dataset will be distributed or maintained over time.

**Additional Feedback:**

Thank you for the response. Unfortunately, given that the data is still being licensed and the documentation still being developed, I don't think this dataset is ripe for publication yet.

**Documentation:**

This is an area where the paper really suffers. The availability and maintenance plans are unspecified in the paper. Is Dropbox the ultimate form in which this data will be distributed? How will the authors make the data discoverable, or ensure availability over time? The included readme is woefully thin, and I'm not seeing code or notebooks to reproduce the results in the paper, beyond some difficult to understand shell scripts.

**Ethics:**

No concerns.

**Relation To Prior Work:**

The citations seem adequate but I'm not an expert in this domain.

**Summary And Contributions:**

This paper presents a large dataset drawn from meical corpora of drug interaction triples. The authors evaluate several state of the art methods for triple extraction and find that their performance is not great, suggesting that more work is needed to improve model performance on this task.

---

> ### Author Response · Authors · 2021-07-14
> **Response**
>
> 1. About dataset distribution.
>
> Although the entire dataset has obtained the necessary license, it is still in the process of our internal review.
> We will release the dataset along with detailed documentation on GitHub after the review process and ensure maintenance.
>
> 2. "scripts that lack documentation or explanation."
>
> The experiment code with detailed documentation has been updated at https://github.com/bert-nmt/BERT-DTI

---

### Decision · Program_Chairs · 2021-07-26

**Decision:**

Reject

**Comment:**

The reviewers feel that the task is important, and the dataset may contain richer information than others. However, the contributions are not sufficient for acceptance, mostly because the authors failed to release the data before the submission. The reviewers thus could not access and check the dataset.  While in response, the authors released data for the review purpose, it might be too late. In addition, two reviewers feel that the dataset is not well described, and some flaws in domain knowledge make the quality of manual checking questionable. A comparison between ChemProt and DK-DTI might also be helpful.